# Influence of Seasonality and Culture Stage of Farmed Nile Tilapia (*Oreochromis niloticus*) with Monogenean Parasitic Infection

**DOI:** 10.3390/ani13091525

**Published:** 2023-05-02

**Authors:** Elisabeth de Aguiar Bertaglia, William Eduardo Furtado, Ângela Teresa Silva e Souza, Manoela Clemente Fernandes, Scheila Anelise Pereira, Elenice Martins Brasil, José Luiz Pedreira Mouriño, Gabriela Tomas Jerônimo, Maurício Laterça Martins

**Affiliations:** 1AQUOS Laboratory Health of Aquatic Organisms, Department of Aquaculture, Federal University of Santa Catarina (UFSC), Rod. Admar Gonzaga 1346, Florianópolis 88040-900, SC, Brazil; 2Department of Infectious Diseases and Public Health, Jockey Club College of Veterinary Medicine and Life Sciences, City University of Hong Kong, Kowloon, Hong Kong, China; 3Programa de Pós-Graduação em Ciências Biológicas, Departamento de Biologia Animal e Vegetal, Universidade Estadual de Londrina (UEL), Londrina 86039-000, PR, Brazil

**Keywords:** aquaculture, ectoparasites, season, net cages, abiotic factors, infection dynamics

## Abstract

**Simple Summary:**

Monogeneans are problematic parasites in fish production because of their epizootic effects and difficulty in administering control/treatment in the production environment. Understanding how abiotic and biotic factors influence the dynamics of these parasites is the first step towards developing efficient strategies for the management of monogeneans in intensive fish farming. This study showed that the highest parasitic indices of monogeneans in Nile tilapia (*Oreochromis niloticus*) were found in the coldest seasons with less precipitation. Furthermore, we showed that there was a clear positive correlation between the developmental culture stages of tilapia and the incidence of monogenean parasitism; that is, the larger the fish, the greater the degree of parasitism. In conclusion, we recommend the adoption of good management practices throughout breeding, especially in seasons that favor parasitism, as a promising strategy for reducing the mortality and economic losses associated with tilapia production.

**Abstract:**

The aim of this study was to observe how abiotic and biotic factors in a tropical region influence the rate of monogenean parasitism in Nile tilapia (*Oreochromis niloticus*) that are farmed in net cages. A total of 240 sexually reversed fish were analyzed, and 20 from each culture stage were collected during each sampling month. Overall, 60 fish were sampled in April (autumn), 60 in August (winter), 60 in November (spring), and 60 in February (summer). Fish were collected from a commercial fish farm located in Capivara Reservoir in the lower Paranapanema River region of Paraná, Brazil. In total, 3290 monogenean parasites were collected from fish gills of the following species: *Cichlidogyrushalli*, *C. thurstonae*, *Scutogyruslongicornis*, *C. cirratus*, *C. sclerosus*, and *C. tilapiae*. Higher parasitological indices were observed in colder seasons with lower precipitation. Autumn had the highest parasitic infection values compared to the other seasons. The occurrence of monogenean parasites showed a negative correlation with season, in contrast to the culture stage, in which there was a positive correlation. These results may provide a means for establishing adequate fish farm management to predict periods of high monogenean infestation.

## 1. Introduction

Nile tilapia (*Oreochromis niloticus*) is the main fish species raised in the net tanks of freshwater reservoirs in Brazil [1]. It is a suitable host for various Monogenea parasites, including those belonging to the family Ancyrocephalidae (Bychowsky, 1937) [2,3,4]. On the African continent, this family of ectoparasites is the most common and includes species that infest the gills of Nile tilapia, which are grouped into two genera: *Cichlidogyrus* Paperna, 1960 and *Scutogyrus* Pariselle and Euzet, 1995 [4,5]. Monogenean parasites and their population dynamics favor high temperatures, high host densities, and good water quality [6,7]. Over a temperature range of 20–28 °C, *Cichlidogyrus* eggs hatch within 2–6 days, releasing free-swimming oncomiracidia (infection phase). They mature within 4–6 days and attach to their hosts, where they live for up to 40 days [8]. Under intensive farming conditions, these parasites exploit various stressors that affect their hosts [5,9,10]. Furthermore, their presence can result in fish mortality and economic losses through primary or secondary infections [11,12,13].

Several biotic and abiotic factors affect monogenean parasitism [14]. The host and its immune response play important roles as components of the innate immune system (complement, lectins, and macrophages) that can bind to monogeneans and cause serious damage to parasites, thus preventing infection [15,16,17,18]. However, in a farming system, the high density of fish and management practices to which they are subjected impose stress on these fish; this makes them more prone to infection by pathogens owing to the weakening of their immune responses [19]. In addition, water quality parameters and rainfall frequency affect parasite abundance by directly affecting the monogenean ectoparasite life cycle, where high temperatures favor egg hatching and the dynamics of the parasitic population. In contrast, larger volumes of water caused by rain can hinder both swimming and arrival of oncomiracidia in their hosts [8,9,20,21,22,23]. The relationship between the size and sex of Nile tilapia and infection rates of monogenean parasites could also be established [17,24,25]. According to Felix-Cuencas et al. [26], there are three culture stages for Nile tilapia: fingerlings, juveniles, and adults. Host size is also an indicator of parasite vulnerability [27].

To the best of our knowledge, most studies investigating the relationship between abiotic and biotic factors and parasitological indices in monogeneans have been conducted in temperate zones, which have very different climate dynamics from tropical zones. Furthermore, it is unclear whether water quality, season, and other factors influence the dynamics of monogenean parasitic infections in tropical regions, [9,25] as reports are scarce and often conflicting. Additionally, the adverse effects of modifying the host habitat from natural to confined can affect the distribution and transmission of parasites [19,20,21]. Parasitological indices are essential tools for aquatic health studies. Understanding parasite biology is important for establishing adequate management and sanitary control strategies, particularly for fish farms [7,10,13]. Monogenean species have short and direct life cycles [28]; therefore, their prevalence, intensity, and average abundance are sensitive to the prevailing biotic and abiotic factors [25].

Thus, the objective of the present study was to observe how abiotic (seasonality) and biotic (culture stages) factors influence monogenean parasitism in Nile tilapia farmed in offshore net cages in tropical regions. This was studiedto predict possible peaks of parasitic infections and to avoid economic losses forfish farms.

## 2. Material and Methods

### 2.1. Fish Collection and Analysis (Biotic Factors)

A total of 240 fish sexually reversed to males were analyzed, and 20 fish from each culture stage in each season were collected in each sampling month, totaling 12 tanks.

Thus, 60 fish were sampled each in April (autumn), August (winter), November (spring), and February (summer). Each month the collection was divided into 3 stages of host development: 80 fingerlings, 80 juveniles, and 80 adults, as standardized by Félix-Cuencas et al. [26].

The stages of development were established from the measurements, weight, and fish farming definitions in the different cages. Fingerlings were fish that had newly been placed in net cages, with mean measurements of 10.77 ± 1.53 cm and 23.85 ± 10.29 g. Juveniles were those with measurements of 22.66 ± 3.01 cm and 247.89 ± 110.67 g. The adults were those that were ready for harvesting, with measurements of 27.53 ± 2.35 cm and 436.49 ± 124.63 g.

The fish were collected from a fish farm located in the Capivara Reservoir, in the region of the lower Paranapanema River in Paraná, Brazil (22°47′22.36″ S; 51°17′46.53″ W). After the harvesting procedure was performed by the farmers, the fish were transported in boxes with ice to the Laboratory of the Fish Farming Station of the State University of Londrina (LEPUEL) for biometric evaluation (standard length in cm and weight in g) and parasitological analysis. After the arrival of the dead fish in the lab, they were divided into groups of fingerlings, juveniles, and adults to assess the “wellbeing” of the fish. The value of the relative condition factor (Kn = Wt/We) was determined individually through the quotient between the empirically registered weight (Wt) and the theoretically expected weight (We) for a given length [29].

### 2.2. Collection and Analysis of Abiotic Factors

The water quality parameters were measured in situ at the time of each collection. The concentration of dissolved oxygen (mg L^−1^) and temperature (°C) were measured in situ, using an oximeter (model Y55) and a thermistor, respectively. Water samples were then collected to determine the following: pH, using a Quimis bench electronic pH meter; total alkalinity (mg L^−1^), by means of titration with diluted sulfuric acid; and nitrite concentration, expressed in mg L^−1^, using the classic spectrophotometric method based on the Griess reaction. Data on total rainfall, from the previous week to the day of collection (totaling 8 days of analysis), were provided by the Instituto das Águas do Paraná.

### 2.3. Parasitological Analysis

The gill arches were removed, and examined with the aid of a stereomicroscope [11] The monogenean parasites found were fixed in A.F.A solution (acetic acid, formaldehyde, and 70% ethyl alcohol) and preserved in 70% alcohol [9]. Parasitological quantification was performed with the aid of a magnifying glass (Zeiss^®^ Stemi dv4, Oberkochen, Germany) in which all the collected specimens were mounted between slide and coverslip, in the ventral position in Hoyer’s medium [28], and subsequently identified with the aid of the DIC microscope (Differential Interference Contrast) model Axio Imager A2 (Zeiss^®^). The evaluation of the sclerotized structures of the specimens found was performed based on previous descriptions [2,4,30,31,32,33]. The prevalence (P%), mean intensity of infestation (MI), and mean abundance (MA) indices were calculated for both the total number of parasites and for each species [34]. Each species collected wasdeposited in the Helminthological Collection of the Oswaldo Cruz Institute (CHIOC), Rio de Janeiro, Brazil (CHIOC 39278, 39279, 39280, 39281, 39282, 39283, 39284, 39285).

### 2.4. Statistical Analysis

All the results were previously submitted to the Shapiro–Wilk and Bartlett tests to verify the normality and homogeneity of variances, respectively. The total quantity of monogenean parasites and each species, with regard to culture stages and seasonality were compared by the non-parametric Kruskal–Wallis test followed by Dunn’s test for comparison of means. Principal Component analysis (PCA) and Spearman’s correlation were made by PAST 4.03 [29]. In Spearman’s correlation, the “X” indicated there were no differences in significance. The numbers of monogenean parasites in total and of each species were also compared with regard to culture stages and months using the Quantitative Parasitology software, which is available online, through bootstrap t tests with 1000 replicates and 95% confidence intervals [30]. For this test, the software Statistica^®^ 10.0 (Statsoft Inc., Tulsa, OK, USA) was used. All tests were performed at a significance level of 5%.

## 3. Results

Among the 3290 monogenean specimens collected from the gills of Nile tilapia, 6species were identified. *Cichlidogyrus halli* Price and Kirk (1967) was the most prevalent species (46.81%) throughout the study period. This was followed by *C. thurstonae* Ergens, 1981 (40.45%), *Scutogyruslongicornis* Paperna and Thurston, 1969 (34.54%), *C. cirratus* Paperna, 1964 (26.36%), *C. sclerosus* Paperna and Thurston, 1969 (15.45%), and *C. tilapiae* Paperna, 1960 (2.27%) (Figure 1). Analyses of the relationships among monogenean species, seasonality, and tilapia culture stages were performed.

The prevalence, intensity, and average abundance indices of the monogenean species are shown in Table 1. Higher rates of parasitism were observed during colder seasons with lower rainfall. Autumn (with a total rainfall of 8.9 mm) had the highest values for the parasitic indices and was significantly different (*p* < 0.05) from that of the other months, followed by winter (0 mm rainfall). Summer (69.6 mm rainfall) was the season with the lowest parasitic infection values. Thus, a negative correlation (r = −0.3297) was recorded between monogenean parasites and the seasons and a positive correlation (r = 0.0069) with the culture stage (Figure 2).

Regarding the culture stages, the prevalence and mean abundance of parasitism showed a significant difference (*p* < 0.05) in adult fish compared with those of intermediate and initial juveniles (Table 2).

In the principal components analysis (PCA) (Figure 3), the results clearly showed a stronger relationship between the total number of monogeneans and dissolved oxygen, alkalinity, and pH parameters, coinciding with lower temperatures. In contrast, warmer stations (spring and summer) had higher temperatures, higher levels of ammonia concentrations, and rainfall. These data are also supported by Spearman’s correlation presented in Figure 2.

Six monogenean species were distributed throughout all seasons. For most species, except *C. cirratus* and *C. tilapiae*, the highest parasitic rates were found in autumn (8.9 mm rainfall), which was significantly different (*p* < 0.05) from those of the other seasons. The most prevalent species in the different seasons were *C. halli* (78.3%) in autumn, *C. cirratus* (47.5%) in winter, *C. halli* (41.7%) in spring, and *C. halli* (21.7%) in summer. *C. tilapiae* was the least prevalent in all seasons evaluated.

All species were found in all tilapia culture stages, except *C. cirratus*, which was not found in the fingerlings (Table 2). *C. halli* and *C. cirratus* had the highest prevalence rates among adult fish, which were significantly different from the rates among fingerlings and juveniles. The prevalence and mean abundance indices of *C. sclerosus* were highest in adult fish and were only significantly different from those of juveniles. The same was observed for *C. thurstonae*, but only for the average intensity of infection. *S. longicornis* had the highest parasitic rates in both adults and juveniles, with a significant difference compared to juveniles (Table 2).

The most prevalent species at each culture stage was *C. halli* (33.3%) among the fingerlings, *C. thurstonae* (40.0%) among the juveniles, and *C. halli* (71.2%) among the adults. No significant differences or correlations were found between the condition factor (Kn) and abundance of each monogenean species (Table 2).

## 4. Discussion

All six species identified in the present study have already been reported to parasitize the gills of Nile tilapia in the northern, southeastern, and southern regions of Brazil [9,31,32,33,35,36,37,38,39,40]. The samples for the present study were collected from the southern region of Brazil, a tropical zone with a humid tropical savanna climate. In this climate, the rainiest months coincide with spring and summer (September–March) and the drought months coincide with autumn and winter (April–September) [41].

Colder months with lower rainfall corresponded with higher parasitic index values, thus establishing a negative correlation between monogenean parasitism and the warmer seasons of the year. Several studies have reported that, in temperate regions, egg hatching, survival, infection capacity, population dynamics, and life expectancy of monogenean parasites depend on high temperatures [8,14,24,42,43]. The temperature variation between the seasons in tropical zones is not as significant as in temperate zones; consequently, high temperatures are recorded nearly all year round [44], which werebetween 19.4 and 28 °C.

In tropical regions, many studies have corroborated that parasitism increases at higher temperatures, such as during spring and summer in Brazil [9,14,39]. However, some authors have observed increased rates of parasitism at lower temperatures [23,41,45]. Recently, researchers [41] observed a similar parasitic community in the same river for the same duration as that in the current study. In addition, it presented higher mean values for prevalence and abundance in autumn and spring, and lower values in winter and summer. These results emphasize the possible influence of seasonality on variation in the infection dynamics of the monogenean community. Akol et al. [23] observed peak levels of *C. tilapiae* and *C. sclerosus* during the dry months of June and August (winter) in Nile tilapia farmed in net tanks in Uganda. Aguirre-Fey et al. [46] showed that, in four species of tilapia grown in Veracruz, Mexico, the abundances of *C. sclerosus*, *C. dossoui*, and *Scutogyrus* sp. were significantly negatively associated with higher temperatures during the survey months. This may be related to an increase in host immune response, which is positively influenced by higher temperatures.

The low rainfall factor observed in the present study is concordant with findings from previous studies, in which precipitation and associated hydrological changes, such as floods and water currents, were found to play an important role in monogenean infection patterns [45]. *O. niloticus* specimens raised in net tanks in Uganda and collected during the dry season harbored greater numbers of *C. tilapiae* and *C. sclerosus* than those collected during the rainy season [23]. Rainfall possibly decreases the transmission of oncomiracidia, which are free-swimming, as the increased volume of water produces currents that change both habitat size and water quality parameters [23]. In contrast, during dry periods, the contraction of habitat size leads to the agglomeration of hosts, which increases their availability and proximity to oncomiracidia, the infective stage of the parasite. Egg hatching is stimulated in the presence of fish mucus [16]. It has also been suggested that monogenean eggs hatch in the gills of infected and immunosuppressed fish, thereby contributing to the success of parasitism in environments with less water [8,11].

We believe that the influence of rainfall is as significant as that of temperature and other water quality parameters. Therefore, it is necessary to examine the relationship between these factors. The season that promoted the greatest success of parasitism was autumn, when the temperature reached 24.3 °C and the rainfall was 8.9 mm. This temperature is in the optimal average temperature range for monogeneans, and, in association with low rainfall, leads to greater parasitism success [14,23]. However, in winter, the temperature of 19.4 °C, which is below the optimal temperature range for egg hatching and parasite propagation, but which occurred in association with no rainfall, had the second highest gross values for parasitic indices. Nevertheless, there was no statistical difference between these indices and those from autumn. The lowest gross values and parasitic indices were obtained in summer, which we believe was due to the high rainfall, although this was associated with the highest temperature. This emphasizes the importance of the host immune response, which is higher in warmer months, in association with the large volume of water caused by rain, which reduces the transmission of oncomiracidia.

In addition to temperature, we observed that the hottest seasons and nitrogenous compounds were positively correlated. Furthermore, these conditions had the lowest parasitic indices. These findings corroborate those of Ácosta-Pérez et al. [7], who analyzed the influence of physicochemical water quality on the parasitic biodiversity of juvenile farmed tilapia at Valle del Mezquital in Mexico and observed that monogeneans of the genus Dactylogyrus showed negative correlations with nitrogenous compounds. In the current study, this may have occurred because higher levels of rainfall provided greater water flow during this period, thereby promoting greater circulation and renewal of water in the cages, and reducing parasitism success. Therefore, the dispersion of parasites may have been a result of the water flow and pressure [47].

Upon analysis of both monogeneans in general and among their species (except for *C. tilapiae*), the highest rate of parasitism was found among adults. Contrary results were found by Akoll et al. [23], who found that the larger the Nile tilapia, the lower the number of monogenean parasites. The results of the present study corroborate the findings of Ibrahim [48] who found a positive correlation between species richness and prevalence and the size of red tilapia *T. zillii*. Lamková et al. [19] also saw this in *Squalius cephalus*. This suggests that the development of the gonads is associated with high levels of steroid hormones, which can suppress immune function and increase the risk of the host becoming more susceptible to infection by parasites [19]. In general, larger fish are more likely to be sexually mature and exposed to a greater diversity or parasitic species for longer periods than smaller fish [49,50,51].

Additionally, larger fish resulted in a lower carrying capacity of the cultivation units because they had the same dimensions in all phases of cultivation, which was due to the greater population density of larger fish. This was also observed by Suliman and Al-Harbi [52] in Nile tilapia reared in Saudi Arabia, who found no correlation between stocking density and parasite occurrence. Unlike several other diseases, conditions associated with high stocking densities are favorable for the horizontal transmission of parasites [53].

Thus, a complex association between abiotic and biotic factors underlies the different interactions between host fish and monogenetic parasites. We drew attention to the fact that several other factors may have affected the hosts and parasites that could not be correlated. Therefore, these results reflect the unique and specific conditions of this environment, that is, the fish and parasites analyzed under farmed conditions. In addition to temperature, increases in other parameters, such as photoperiod, pH, salinity, oxygen, and ammonia concentration, generally increase the immune response of fish [54], and some of these indices varied over the course of this study. The same applies to parasites because, in addition to temperature, cycles among monogeneans are also affected by other factors, such as habitat type, [23] water salinity, [55] eutrophication, [56] host sex, age, size, and ecology [23,24,57].

Notably, the abiotic data presented in this study reflect only the time of biological material collection and do not correspond to a complete daily or weekly series. However, it was considered using the seasons since they are well-defined in southern Brazil where the fish farm in this study was located. In future studies, we suggest that companies collect extensive and complete environmental data regularly to further correlate the results.

## 5. Conclusions

Based on these results, rainfall combined with water quality parameters influenced monogenean parasitism in Nile tilapia, and developmental stage also influenced this parasitism. However, these results only correspond to the time of collection in the conditions of the fish farm analyzed. Despite this limitation, we infer that parasitic indices can be used to assist in the development of management strategies and adequate sanitary control measures for fish farms.

## Figures and Tables

**Figure 1 animals-13-01525-f001:**
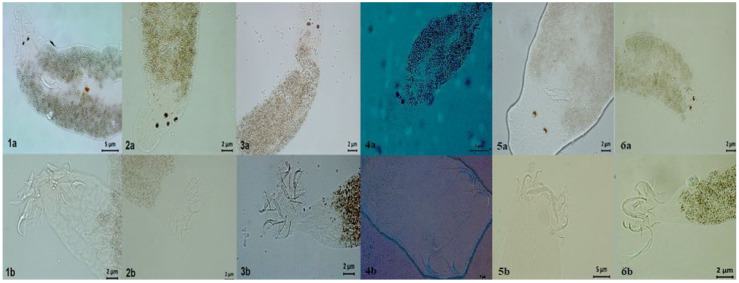
Monogenean specimens identified through the visualization of the male copulatory complex (**a**) and haptor structures (**b**). (1) *Cichlidogyrus halli*; (2) *C. thurstonae*; (3) *Scutogyrus longicornis*; (4) *C. cirratus*; (5) *C. sclerosus*; (6) *C. tilapiae*.

**Figure 2 animals-13-01525-f002:**
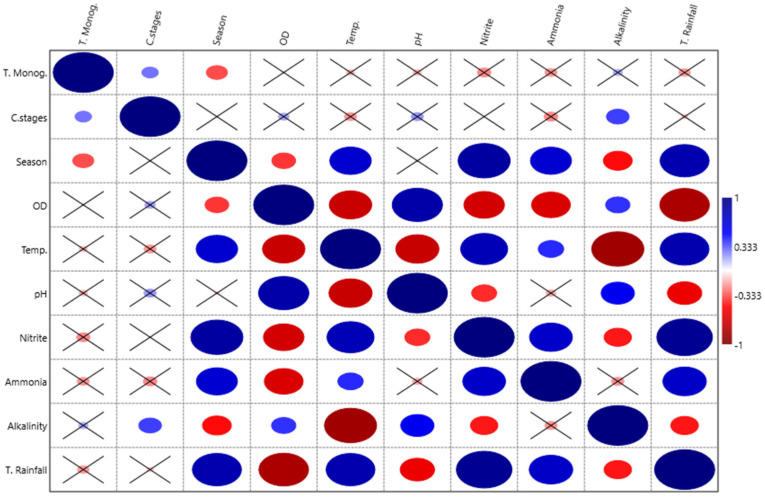
Spearman’s correlation showing the abiotic factors that were related to the total number of Monogenea observed (blue: positive correlation, red: negative correlation). Squares containing (x) indicate that there was no significant difference (*p* > 0.05).

**Figure 3 animals-13-01525-f003:**
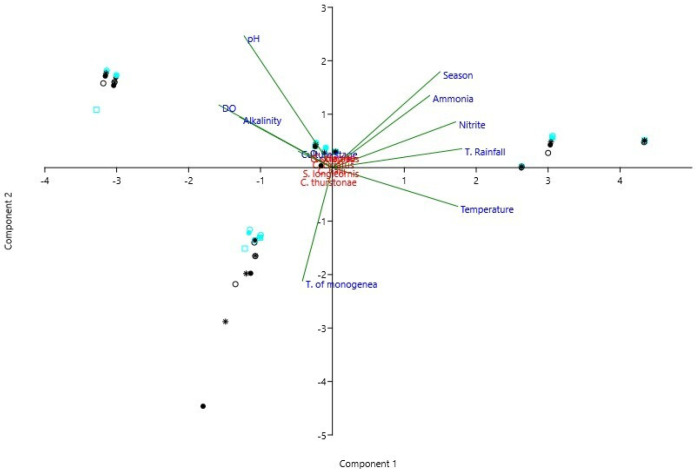
Principal Component Analysis (PCA) using a correlation of matrix where the different symbols show the species of Monogenea in relation to abiotic factors (Dissolved Oxygen-DO, Temperature, pH, Nitrite, Ammonia, Alkalinity, and Total rainfall), season, total Monogenea and culture stage. □ Aqua Square shows *C. cirratus*; ○ Black Circle shows *C. halli*; ● Aqua Dot shows *C. sclerosus*; ● Black Dot shows *C. thurstonae*; ○ Aqua Circle shows *C. tilapiae*; ⁕ Black Star shows *S. longicornis*.

**Table 1 animals-13-01525-t001:** Epizootiological parameters (±SD = standard deviation) of parasitic infestation species individually in Nile tilapia cultured in net cages on the Paranapanema River, PR, Southern Brazil between April (autumn) 2010 and February (summer) 2011. N = total number of parasites; P% = prevalence; MA = mean abundance; MI = mean intensity of infestation *.

Species	Indices	Autumn	Winter	Spring	Summer	*p* Value
*C. sclerosus*	N	28 ^a^	7 ^a^	4 ^a^	10 ^a^	0.0006
	P%	31.7% ^a^	10.0% ^b^	6.7% ^b^	11.7% ^b^	0.0015
	MI ± SD	1.47 ± 0.69	1.75 ± 0.95	1.00 ± 0.00	1.42 ± 0.78	0.422
	MA ± SD	0.46 ± 0.79	0.17 ± 0.59	0.06 ± 0.25	0.16 ± 0.52	0.675
*C. thurstonae*	N	1130 ^a^	77 ^b^	109 ^b^	57 ^b^	0.000
	P%	76.7% ^a^	30.0% ^b^	31.7% ^b^	20.0% ^b^	<0.0001
	MI ± SD	24.57 ± 30.80 ^a^	6.41 ± 7.89 ^b^	5.73 ± 13.21 ^b^	4.75 ± 3.04 ^b^	0.001
	MA ± SD	18.83 ± 28.87 ^a^	1.92 ± 5.14 ^b^	1.81 ± 7.77 ^b^	0.95 ± 2.32 ^b^	0.002
*C. tilapiae*	N	1	0	3	1	0.841
	P%	1.7%	0.0%	3.3%	1.7%	0.9090
	MI ± SD	1.00 ± NA	-	1.50 ± 0.70	1.00 ± NA	-
	MA ± SD	0.01 ± 0.12	0.00 ± 0.00	0.05 ± 0.28	0.01 ± 0.12	0.468
*S. longicornis*	N	558 ^a^	48 ^b^	37 ^b^	33 ^b^	0.0000
	P%	65.0% ^a^	32.5% ^b^	25.0% ^b^	15.0% ^b^	<0.0001
	MI ± SD	14.31 ± 13.12 ^a^	3.69 ± 2.78 ^b^	2.46 ± 1.40 ^b^	3.66 ± 2.91 ^b^	0.001
	MA ± SD	9.30 ± 12.58 ^a^	1.20 ± 2.33 ^b^	0.61 ± 1.27 ^b^	0.55 ± 1.70 ^b^	0.001
*C. halli*	N	529 ^a^	96 ^ab^	112 ^ab^	94 ^b^	0.0013
	P%	78.3% ^a^	45.0% ^b^	41.7% ^b^	21.7% ^b^	<0.0001
	MI ± SD	11.26 ± 12.43 ^a^	5.33 ± 5.47 ^b^	4.48 ± 7.50 ^b^	7.23 ± 8.35 ^b^	0.028
	MA ± SD	8.81 ± 11.93 ^a^	2.40 ± 4.50 ^b^	1.86 ± 5.27 ^b^	1.56 ± 4.82 ^b^	0.001
*C. cirratus*	N	106 ^a^	200 ^a^	41 ^ab^	9 ^b^	0.0000
	P%	30.0% ^b^	47.5% ^a^	25.0% ^b^	10.0% ^b^	0.0004
	MI ± SD	5.88 ± 5.02	10.53 ± 21.36	2.73 ± 2.12	1.50 ± 0.83	0.69
	MA ± SD	1.76 ± 3.83	5.00 ± 15.46	0.68 ± 1.57	0.15 ± 0.51	0.174
Total	N	2352 ^a^	428 ^b^	306 ^bc^	204 ^c^	0.000

* Different letters in the same line indicate a significant difference by the Kruskal–Wallis test (*p* < 0.05) using multiple comparisons of mean ranks for all groups, compared among the season.

**Table 2 animals-13-01525-t002:** Epizootiological parameters ((±SD = standard deviation) of parasitic infestation by species individually in Nile tilapia cultured in net cages on the Paranapanema River, PR, Southern Brazil in different culture stages. N = total number of parasites; Kn = relative condition factor; P% = prevalence; MA = mean abundance; MI = mean intensity of infestation *.

Species	Indices	Fingerling	Juvenile	Adult	*p* Value
*C. sclerosus*	N	1 ^b^	16 ^b^	32 ^a^	0.0003
	Kn	1.00 ± 0.03	1.00 ± 0.02	0.99 ± 0.01	0.9982
	P%	1.7% ^b^	15.0% ^ab^	26.2% ^a^	0.0001
	MI±SD	1.00 ± NA	1.33 ± 0.65	1.52 ± 0.75	-
	MA±SD	0.01 ± 0.12 ^b^	0.20 ± 0.53 ^ab^	0.40 ± 0.77 ^a^	0.007
*C. thurstonae*	N	95 ^b^	333 ^a^	945 ^a^	0.000
	Kn	1.00 ± 0.03	1.00 ± 0.02	0.99 ± 0.01	0.9982
	P%	30.0%	40.0%	48.8%	0.0841
	MI±SD	5.27 ± 3.95 ^b^	10.41 ± 13.36 ^ab^	24.23 ± 33.68 ^a^	0.046
	MA±SD	1.58 ± 3.23	4.16 ± 9.81	11.81 ± 26.35	0.08
*C. tilapiae*	N	1	2	2	0.8418
	Kn	1.00 ± 0.03	1.00 ± 0.02	0.99 ± 0.01	0.9982
	P%	1.7%	1.2%	2.5%	1.000
	MI±SD	1.00 ± NA	2.00 ± NA	1.00 ± 0.00	-
	MA±SD	0.01 ± 0.12	0.02 ± 0.22	0.02 ± 0.15	0.959
*S. longicornis*	N	12	164	500	0.0564
	Kn	1.00 ± 0.03	1.00 ± 0.02	0.99 ± 0.01	0.9982
	P%	11.7% ^b^	40.0% ^a^	46.2% ^a^	<0.0001
	MI±SD	1.71 ± 1.49 ^b^	5.12 ± 6.73 ^a^	13.51 ± 12.98 ^a^	0.016
	MA±SD	0.20 ± 0.73 ^b^	2.05 ± 4.91 ^a^	6.25 ± 11.08 ^a^	0.014
*C. halli*	N	171 ^c^	199 ^b^	461 ^a^	0.0000
	Kn	1.00 ± 0.03	1.00 ± 0.02	0.99 ± 0.01	0.9982
	P%	33.3% ^b^	32.5% ^b^	71.2% ^a^	<0.0001
	MI±SD	8.55 ± 6.33	7.65 ± 10.72	8.08 ± 11.26	0.965
	MA±SD	2.85 ± 5.42	2.48 ± 7.02	5.76 ± 10.17	0.154
*C. cirratus*	N	0 ^b^	49 ^b^	307 ^a^	0.0000
	Kn	1.00 ± 0.03	1.00 ± 0.02	0.99 ± 0.01	0.9982
	P%	0.0% ^b^	22.5% ^b^	50.0% ^a^	<0.0001
	MI±SD	0	2.72 ± 1.87	7.67 ± 15.21	0.146
	MA±SD	0.00 ± 0.00 ^b^	0.61 ± 1.43 ^b^	3.83 ± 11.36 ^a^	0.032
Total	N	280 ^c^	763 ^b^	2247 ^a^	0.000

* Different letters in the same line indicate a significant difference by the Kruskal–Wallis test (*p* < 0.05) compared among development stages.

## Data Availability

The data that support the findings of this study are available upon request from the correspondence author (M.L. Martins).

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
