# Peer review of "Influence of Seasonality and Culture Stage of Farmed Nile Tilapia (Oreochromis niloticus) with Monogenean Parasitic Infection"

_animals, 2023, doi:10.3390/ani13091525_

Round 1
Reviewer 1 Report
The study of correlations between environmental parameters, pathogens and diseases is a field of great interest in aquaculture. The work presented here aims to shed light on these issues, in order to improve the knowledge of the incidence of parasites in freshwater fish production, their impact on production (productivity indicators), and their prevention. Gaining knowledge about these issues is of key importance for a better zootechnical and health management and are needed for better understanding the potential impact of climate change.
.- About collection of biotic factors. Table 1 reports water quality and rainfall parameters during the study period (February 2011 to April 2012). Data about temperature, pH, dissolved oxygen, alkalinity, nitrite, ammonia were collected on just one day, reason why SD are 0 or very low. It is not explained where water parameters where measured. ¿E.g. in different cages of tilapia of different sizes? ¿How many? On the other hand rainfall is a 8 days average provided by the Instituto das Águas do Paraná. IT IS POINTED OUT, this is one of the main criticism, as one day measurement for water parameter and 8 day rainfall cannot not be considered as representative data for a season. My proposal is that authors try to get a better series of data for that period, from the farm records and form the Instituto das Águas do Paraná for the Capivara reservoir.
.- In the manuscript the authors mainly discuss possible correlations of temperature and rainfall with parasite incidence and prevalence, and forget to comment and discuss the other recorded parameters. Some studies have found correlations between parasites in tilapia and different water quality parameters other than temperature and rainfall, e.g. about Dissolved Oxygen. That should be analyzed and discussed.
.- The fact that the results presented here come from data more than 10 years old should not be a problem, as long as the authors: a) Certify to the editor that this research has not been published before and explain why this has not been the case If the work here presented was part of a Master or PhD, may that should be included in Acknowledgements; and b) Contrast and discuss their results with more recent work. Authors should look for and include more recent work in the introduction and discussion. For example, Acosta-Pérez et al. 2022 (Acosta-Pérez, V.-J.; Vega-Sánchez, V.; Fernández-Martínez, T.-E.; Zepeda-Velázquez, A.-P.; Reyes-Rodríguez, N.-E.; Ponce-Noguez, J.-B.; Peláez-Acero, A.; de-la-Rosa-Arana, J.-L.; Gómez-De-Anda, F.-R. Physicochemical Water Quality Influence on the Parasite Biodiversity in Juvenile Tilapia (Oreochromis spp.) Farmed at Valle Del Mezquital in the Central-Eastern Socioeconomic Region of Mexico. Pathogens 2022, 11, 1076. https://doi.org/10.3390/pathogens11101076), where the authors analyse the correlation index between parasite prevalence and physicochemical water quality parameters using a Pearson correlation matrix.
.- The authors should make a more complete statistical analysis to assess the possible existence of correlations between the recorded environmental parameters described in table 1 and parasitosis. Note that there are significant differences in most parameters. In my opinion, different statistical methods can be used to analyze these possible correlations, i.e. Multivariate analysis or Principal Component Analysis (PCA).
.- This critique of statistical analysis connects to the "Data Availability Statement". As a reviewer of this manuscript, I propose that authors make the data available to anyone interested in an repository or as additional material together the manuscript, since providing the data only after request may is more tedious and, may be no possible after few years of publication, if the author has retired or died. Note this manuscript is submitted more than ten years after data was obtained.
.- About the use on Ontogeny and Development stages. In my opinion the use of those terms is not appropriate for this work. Although ontogeny refers to all development stages from the time of fertilization of the egg to adult, it is normally more used for those stages where more changes occurs. In this study, most development stages have been completed when the study starts. Thus, I suggest that authors change the terms and speak about culture stages. Moreover, the differences found between the studied stages (juvenile, intermediate and adult) are more different grow under culture stages. The fact that there are higher prevalence on adults than on juveniles may be due to more causes than those discussed in the manuscript, i.e. the accumulation of parasites during the culture/ongrowing periods, or because the studied parasites may slowly alter the gills mucosa tissue, facilitating parasitosis in following periods. Those possible causes need to be contrasted and discussed too.
.- About the use of “seasons” and “months”. In table 1 and Figure 1, the authors divide the study periods in seasons. However, in table 2 and in the text they refer many times to months. Once that in Material and Methods is explained when the fish collection was made, it’s better to speak only/mainly about seasons to avoid misunderstanding, especially for readers from the North Hemisphere.
.- I propose that tables 2 and 3 present N, P%, MI and MA for Total, as authors frequently discuss results referring to the total monogean parasitic infection.
.- In lines 217-219, it is said tat “In this study, the colder months with lower rainfall corresponded to higher values of parasitism, thus establishing a negative correlation between parasitism by Monogenea and the months.” I am not sure about this statement, as according to table 1, the colder season is winter (19.4ºC), and prevalence is higher in fall (24.3ºC), when temperature is closer to spring (26.5ºC).
Reviewer 2 Report
The ms “Influence of seasonality and development stages of farmed Nile tilapia Oreochromis niloticus on monogenean parasitic infection” by Elisabeth de Aguiar Bertaglia et al. describes the parasitological data obtained by analyzing epizootiological data of fish kept in floating cages during a year in tropical Brazil; the results are interesting from both basic parasitological and applied aquacultural perspectives.
The text is generally well written and flows nicely; however, I highly recommend that the authors carefully revise their use of references, as I detected several instances where the papers that are used to support the statements of the text are not the best choice possible – or worse, are not directly relevant to the topic addressed.
A couple of minor details are highlighted in the annotated version of the text, which I include for the authors to consider during revision.
I would recommend that the authors be invited to resubmit a revised version of the ms.

Round 2
Reviewer 1 Report
The authors have taken into account all the corrections and suggestions made in the first revision. The work is significantly improved in this way. It remains, in my opinion, only one significant improvement to be made. My suggestion implies little additional work, but I consider it significant.
The improvement to be made is related to the quality of the environmental abiotic data, which have been taken only at the time of fish sampling for the parasitism studies. The data are from a single year and point in time, and do not correspond to a complete daily or weekly series. They are what they are, and what was taken at that time. This is very important, and should be discussed (discussion section) properly, to put the scope of the results in accordance with this methodological constrain, whih affects also the extend of the conclusions.
For example, the paper mentions that the study has shown seasonal changes in the parasitism studied, and that correlations have been found with rainfall, usually lower in winter, or that no correlations have been found with other parameters, etc. Thus, the study highlights the need to work on the search for environmental correlations with parasitism and zoo-technical performance, and that requires companies to regularly collect extensive and complete environmental data, as well as zoo-technical and sanitary data.
Best regards
Author Response
Dear Dr. Lyle Zeng,
Thank you for giving us the opportunity to submit a revised version of our paper. We submit a revised draft of the manuscript entitled " Influence of seasonality and culture stage of farmed Nile tilapia Oreochromis niloticus with a monogenean parasitic infection" for publication in Animals. We appreciate the time and effort that you and the reviewers dedicated to providing feedback on our manuscript and are grateful for the insightful comments and valuable improvements to our paper. We have incorporated all the suggestions made by the reviewers. Please see below, in blue, for a point-by-point response to the reviewer’ comments and concerns. Also, all the yellow highlighted comments in the manuscript were revised and the appropriate correction were made.
Yours sincerely,
Dr. Maurício Laterça Martins
Response to Reviewers
Reviewer 1
The authors have taken into account all the corrections and suggestions made in the first revision. The work is significantly improved in this way. It remains, in my opinion, only one significant improvement to be made. My suggestion implies little additional work, but I consider it significant.
We would like to express our deep thanks to reviewer, for the valuable comments and the affirmation to our study.
The improvement to be made is related to the quality of the environmental abiotic data, which have been taken only at the time of fish sampling for the parasitism studies. The data are from a single year and point in time, and do not correspond to a complete daily or weekly series. They are what they are, and what was taken at that time. This is very important, and should be discussed (discussion section) properly, to put the scope of the results in accordance with this methodological constrain, whih affects also the extend of the conclusions.
For example, the paper mentions that the study has shown seasonal changes in the parasitism studied, and that correlations have been found with rainfall, usually lower in winter, or that no correlations have been found with other parameters, etc. Thus, the study highlights the need to work on the search for environmental correlations with parasitism and zoo-technical performance, and that requires companies to regularly collect extensive and complete environmental data, as well as zoo-technical and sanitary data.
We have included a paragraph about this concern. The same can be found in the lines 302-307 (highlighted in yellow).
As for the concern with the correlations with the zootechnical parameters, they are answered in lines 193-204, as well as the discussion presented in lines 273-290. The conclusion has also been reworded, as can be seen in lines 310-315 (highlighted in yellow).
Best regards
Maurício Laterça Martins
